# An Inverse Partial Optimal Transport Framework for Music-guided Movie Trailer Generation

## ABSTRACT

Trailer generation is a challenging video clipping task that aims to select highlighting shots from long videos like movies and reorganize them in an attractive way. In this study, we propose an inverse partial optimal transport (IPOT) framework to achieve music-guided movie trailer generation. In particular, we formulate the trailer generation task as selecting and sorting key movie shots based on audio shots, which involves matching the latent representations across visual and acoustic modalities. We learn a multi-modal latent representation model in the proposed IPOT framework to achieve this aim. In this framework, a two-tower encoder derives the latent representations of movie and music shots, respectively, and an attention-assisted Sinkhorn matching network parameterizes the grounding distance between the shots' latent representations and the distribution of the movie shots. Taking the correspondence between the movie shots and its trailer music shots as the observed optimal transport plan defined on the grounding distances, we learn the model by solving an inverse partial optimal transport problem, leading to a bi-level optimization strategy. We collect real-world movies and their trailers to construct a dataset with abundant label information called CMTD and, accordingly, train and evaluate various automatic trailer generators. Compared with state-of-the-art methods, our IPOT method consistently shows superiority in subjective visual effects and objective quantitative measurements.

## CCS CONCEPTS

• **Computing methodologies → Matching**; **Learning latent representations**; **Video summarization**.

## KEYWORDS

Trailer generation, video clipping, inverse optimal transport, movie-trailer dataset

**ACM Reference Format:**
Anonymous Author(s). 2018. An Inverse Partial Optimal Transport Framework for Music-guided Movie Trailer Generation. In *Proceedings of Make sure to enter the correct conference title from your rights confirmation email (Conference acronym 'XX)*. ACM, New York, NY, USA, 10 pages. https://doi.org/XXXXXXX.XXXXXXX

## 1 INTRODUCTION

As collections of movie highlights that may attract audiences, trailers play a central role in movie promotion. Unlike video summarization [31, 32, 49], which selects key frames or shots to alleviate the redundancy of video content while keeping the completeness of the storyline, trailer generation [19, 33, 40] needs to select attractive movie highlights but reorganize them to hide the original movie's storyline to some extent. The selection and reorganization of movie shots are determined by various factors, e.g., the semantics of background music, the synchronization of visual and acoustic content, the logical flow of characters' dialogues, and so on, which requires a deep understanding of the movie. Therefore, generating a high-quality movie trailer involves sophisticated video clipping and editing, which is time-consuming and labor-intensive (and thus, expensive). Typically, the trailer of a Hollywood blockbuster may require months of work by a team of professional editors to select the movie highlights and align them with background music.

Due to the above fact, many academic and industrial researchers have made efforts to improve the efficiency of movie trailer generation, gradually making the whole process automatic. Currently, some music-guided movie trailer generation methods, especially those learning-based ones [25, 42, 55], have been proposed, which generate trailers from movies automatically based on given background music. At the same time, some commercial software like Muvee [12] is developed to achieve music-guided video clipping and trailer generation. However, when utilizing background music, these methods mainly focus on synchronizing movie shots according to the music rhythm while ignoring the semantic alignment between visual and acoustic information. As a result, the performance of the methods is still unsatisfactory in practical applications. What is worse, the learning-based methods often suffer from the scarcity of labeled training data. For example, the point process-based method in [55] needs to learn an attractiveness model based on the movies with audiences' fixation information collected by professional eye trackers. The emotion correlation-based method in [25] requires video and audio shots to be labeled with manually defined emotion categories. Because such annotation is difficult and time-consuming, the datasets they used are limited in size, leading to a high risk of overfitting.

To overcome the above challenges and boost the performance of automatic trailer generation, in this study, we propose a novel music-guided movie trailer generation method with the help of computational optimal transport techniques. As illustrated in Figure 1, we formulate the music-guided trailer generation task as selecting and sorting key movie shots based on given audio shots and establish an inverse partial optimal transport (IPOT) framework to learn a model to achieve this aim. In particular, given a movie and its corresponding trailer, we first leverage a two-tower encoder to obtain the latent representations of movie shots and trailer music shots, respectively. Given the visual and acoustic latent representations,

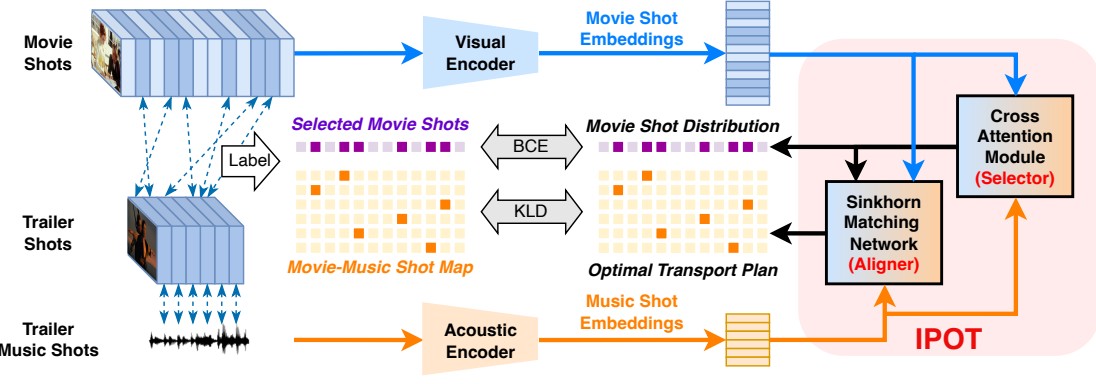

**Figure 1: An illustration of our IPOT framework for learning a music-guided movie trailer generator.**

our IPOT framework applies an attention-assisted Sinkhorn matching network to parameterize the distribution of movie shots and the grounding distances between the latent representation across the two modalities. Accordingly, whether a movie shot should be selected to construct a trailer or not is determined by the movie shot distribution, and the alignment between the movie and music shots is achieved by solving a partial optimal transport problem based on the grounding distances and the movie shot distribution. In the training phase, we learn the model by fitting the movie shot distribution and the cross-modal optimal transport plan to the ground truth movie shot indicator and movie-music shot map, leading to the proposed IPOT framework. A bi-level optimization strategy is applied to learn the model effectively, leading to a movie shot selector and a movie-music shot aligner.

To train our model and compare it with state-of-the-art trailer generators, we collect real-world movies and their official trailers and construct a comprehensive movie-trailer dataset (CMTD) with abundant label information. Compared to existing movie-trailer datasets [11, 18, 47], CMTD contains multiple official trailers for each movie and provides segmentation information for shots in all movies, trailers, and corresponding trailer music. The movie shots and music shots are aligned by matching the movie shots with the corresponding trailer shots. In order to adapt to different tasks and future studies, we also provide the metadata related to the movies, such as subtitles, synopsis, turning points annotations, and so on. To the best of our knowledge, CMTD might to be largest labeled movie-trailer dataset at the current stage.

In summary, the contributions of this work include two folds:

- We propose a novel and effective IPOT framework for music-guided movie trailer generation. Applying the proposed learning framework leads to a new optimal transport-based solution to music-guided movie trailer generation task.
- We construct a new public[1] comprehensive movie-trailer dataset for movie trailer generation and future video understanding tasks. We train and evaluate various trailer generators on the dataset. Experimental results demonstrate the superiority of our IPOT-based trailer generator on both objective measurements and subjective effects.

---

[1]We will release the dataset after acceptance. The trade-off between research acceleration and intellectual property protection is discussed in Section 4.2.

## 2 RELATED WORK

### 2.1 Video understanding and trailer generation

Video understanding is an extensive research field that involves exploring the semantics of video content and aligning it with other modalities, such as texts and audio. As typical video understanding tasks, video retrieval [7, 10] aims to search videos according to their relevance to textual queries, and video tagging [29, 48] and temporal action localization [26, 58, 60] aim to annotate videos or the scenes in them automatically. The development of deep learning further triggers the studies of complicated video understanding tasks, such as video captioning (i.e., generating textual descriptions of videos) [13, 45], video question answering (VQA, i.e., answering questions based on given videos) [22, 59], and video summarization and storytelling [31, 32, 49]. Recently, many video generation methods have been proposed and achieved encouraging performance, e.g., SORA and other related work [17, 39], indicating that the GPT-driven generative models may own strong capability of video semantic understanding in the human level.

As one of the most challenging video understanding tasks, trailer generation selects impressive shots based on understanding the video content and reorganizes the selected shots in an attractive way. The early methods mainly select and sort video shots based on the utilization of side information. To our knowledge, the work in [28] proposes the first user attention model for trailer generation. Early work in [19, 40] intends to identify impressive audio-visual components by affective content analysis to aid in trailer generation. The work in [1] selects trailer moments in soap operas by combining visual and dialogue information. With the development of machine learning techniques, some learning-based methods are proposed to generate trailers under the guidance of music and texts. The work in [55] considers background music in trailer generation and presents a visual attractiveness model based on point process theory. The work in [25] depends on emotion categories to align images, text, and audio in latent space, selecting and reorganizing video shots by maximizing emotion score.

However, the above methods do not consider the semantic consistency between visual and acoustic embeddings, and they heavily rely on videos with detailed annotations or side information, such as frame-level fixation scores, manually defined emotion labels,

and so on. However, the current movie-trailer datasets [11, 18, 47] are far from satisfactory. They neither contain multiple trailers corresponding to one movie to achieve conditional learning nor have fine-grained annotations for useful supervision. As a result, the current learning-based trailer generators often suffer from severe overfitting issues. Motivated by the scarcity of high-quality data, we built a movie-trailer dataset with detailed annotations and abundant metadata in this study.

## 2.2 Optimal transport for matching

As a valid metric of probability measures, optimal transport (OT) [44] has been widely used for distribution comparison and matching. In particular, given two distributions defined in a sample space, we can measure their discrepancy and infer the correspondence between their samples by solving an optimal transport problem and deriving an optimal transport plan (or called coupling [44]) between them accordingly. Therefore, many machine learning tasks that involve matching problems can be modeled as optimal transport problems, e.g., domain adaptation [2, 56], graph matching [4, 36, 54], point cloud registration [37, 38], cross-modal alignment [3, 15, 23], and so on, which all achieve promising results. Typically, the OT problem aims to derive an optimal transport plan to minimize the transport costs between two distributions, which corresponds to a linear programming problem. To solve the problem efficiently (and approximately), Sinkhorn-scaling algorithm [6], proximal point method [51], and Bregman alternating direction method of multipliers (BADMM) [26, 46, 52] are proposed and greatly alleviate the computational complexity of the problem.

Recently, inverse optimal transport (IOT) has been proposed, which aims to optimize the grounding distances associated with samples or their latent representations given observed optimal transport plans [5, 24, 41]. The IOT problem leads to a new learning paradigm to solve a set of latent representation and matching problems, which has been used in many applications. The work in [57] proposes an IOT-based model called IOT-Match for legal case matching, which can generate natural language explanations for matched legal cases and is robust to label insufficiency. The work in [49] proposes to learn a projection layer to achieve semantic alignment between visual and textual representations via IOT techniques in a self-supervised setting. Typically, the IOT problem corresponds to a bi-level optimization problem, which can be solved effectively by the hypergradient method in [27, 50]. In this study, the correspondence between movies and trailers in our dataset can naturally be seen as observed OT plans, which motivates us to propose the inverse partial optimal transport framework for music-guided trailer generation.

# 3 PROPOSED METHOD

## 3.1 Problem statement and modeling principle

Suppose that we have a set of movies and their corresponding trailers, denoted as $\mathcal{D} = \{(\mathcal{M}_n, \mathcal{V}_n, \mathcal{A}_n, T_n)\}_{n=1}^N$. Here, $\mathcal{M}_n = \{m_{i,n}\}_{i=1}^{I_n}$ represents the $I_n$ shots of the $n$-th movie, which corresponds to different scenes happening in the movie. $\mathcal{V}_n = \{v_{j,n}\}_{j=1}^{J_n}$ and $\mathcal{A}_n = \{a_{j,n}\}_{j=1}^{J_n}$ represents the $n$-th trailer, which contains $J_n$ video and audio shots segmented according to the timestamps of different scenes. In general, the trailer shots are selected from the movie, so we can construct an alignment matrix $\mathcal{T}_n = [t_{ij,n}] \in \{0,1\}^{I_n \times J_n}$, where $t_{ij,n} = 1$ indicates that the $j$-th trailer shot corresponds to the $i$-th movie shot. Obviously, this alignment matrix $T_n$ also works as a movie-music shot map, providing the correspondence across the visual and acoustic modalities, and accordingly, $\mu_n = T_n \mathbf{1}_{J_n} \in \{0,1\}^{I_n}$ indicates which movie shots are selected to generate the trailer.

In this study, given a movie $\mathcal{M}$ and a piece of music $\mathcal{A}$, we aim to generate a trailer $\mathcal{V}$ for the movie. We can formulate this music-guided movie trailer generation task as selecting and sorting movie shots conditioned on the music shots, which corresponds to predicting the alignment matrix $T$ between $\mathcal{M}$ and $\mathcal{A}$ (and the associated movie shot indicator $\mu$). In the following content, we will show that when the above dataset is available, we can learn a multi-modal representation and matching model (as illustrated in Figure 1) in a supervised way to achieve this aim, leading to the proposed inverse partial optimal transport framework.

## 3.2 Model architecture

### 3.2.1 Multi-modal Self-attentive latent representation.
In this study, for an arbitrary movie with $I$ shots and its corresponding trailer with $J$ shots, we first apply the pretrained ImageBind [14] to extract initial embeddings of the movie shots and trailer music shots, respectively, i.e., $M = f_M(\mathcal{M}) = [m_i] \in \mathbb{R}^{I \times D}$ and $A = f_A(\mathcal{A}) = [a_j] \in \mathbb{R}^{J \times D}$. To make the embedding adaptive to our task and take the temporal correlation of the shots into account, we pass the embeddings of the two modalities through two multi-layer perceptrons (MLPs) and further encode each modalities' embedding by two self-attention (SA) modules, i.e.,

$$M' = \mathrm{MLP}_M(M),\ A' = \mathrm{MLP}_A(A),$$
$$M^s = \mathrm{Softmax}\Big(\frac{(M'W_1^m)(M'W_2^m)^\top}{\sqrt{D}}\Big)M'W_3^m, \qquad (1)$$
$$A^s = \mathrm{Softmax}\Big(\frac{(A'W_1^a)(A'W_2^a)^\top}{\sqrt{D}}\Big)A'W_3^a,$$

where $\mathrm{MLP}_M, \mathrm{MLP}_A : \mathbb{R}^D \mapsto \mathbb{R}^D$, $\{W_i^m, W_i^a \in \mathbb{R}^{D \times D}\}_{i=1}^3$, and $M^s = [m_i^s] \in \mathbb{R}^{I \times D}$ and $A^s = [a_j^s] \in \mathbb{R}^{J \times D}$ are the proposed latent representations of movie shots and trailer music shots, respectively. The MLPs together with the self-attention modules lead to a multi-modal encoder with a two-tower architecture. As aforementioned, based on the latent representations, we would like to select key movie shots and align them with the trailer music shots, which is achieved by the following two modules.

### 3.2.2 Cross-attention movie shot selector.
Our trailer generator needs to select key movie shots conditioned on given music. Therefore, we propose a cross-attention movie shot selector to predict which movie shots should be selected. In particular, we utilize a cross-attention (CA) module to capture the interactions between the visual and acoustic latent representations, i.e.,

$$\bar{M} = M^s + \mathrm{Softmax}\Big(\frac{(M^s W_4^m)(A^s W_4^a)^\top}{\sqrt{D}}\Big)A^s W_5^a,$$
$$\bar{A} = A^s + \mathrm{Softmax}\Big(\frac{(A^s W_5^m)(M^s W_6^a)^\top}{\sqrt{D}}\Big)M^s W_6^m, \qquad (2)$$

---

**Algorithm 1** SinkhornNet($D, \frac{1}{\|\hat{\mu}\|_1}\hat{\mu}, \gamma; \lambda$)

---

1: Initialize $K = \exp(-D/\lambda)$ and $a = 1$.
2: **While** not converge **do**
3: $\quad b \leftarrow \frac{\gamma}{K^\top a}$ and then $a \leftarrow \frac{\hat{\mu}}{IKb}$.
4: **return** $\widehat{T} = \text{diag}(a)K\text{diag}(b)$

---

where $\{W_i^m, W_i^a \in \mathbb{R}^{D \times D}\}_{i=4}^6$, $\bar{M} = [\bar{m}_i] \in \mathbb{R}^{I \times D}$ and $\bar{A} = [\bar{a}_i] \in \mathbb{R}^{J \times D}$ are the final latent representations of the two modalities. Passing $\bar{M}$ through the following MLP results in a vector indicating the probabilities of selecting different movie shots, i.e.,

$$\hat{\mu} = [\hat{\mu}_i] = \text{Sigmoid}(\text{MLP}(\bar{M})) \in [0,1]^I, \qquad (3)$$

where each $\hat{\mu}_i$ indicates the probability that the $i$-th movie shot is selected to generate a trailer.

*3.2.3 Sinkhorn-based movie-music aligner.* Besides selecting key movie shots, we need to determine the order of the selected movie shots and make them aligned to the music shots. In this study, we propose a Sinkhorn matching network as the movie-music aligner. In particular, given the visual and acoustic latent representations $\bar{M}$ and $\bar{A}$, we can construct a distance matrix $D = [d(\bar{m}_i, \bar{a}_j)] \in \mathbb{R}^{I \times J}$, whose element $d(\bar{m}_i, \bar{a}_j)$ represents the Euclidean distance between the latent representation of the $i$-th movie shot and that of the $j$-th music shot. The Sinkhorn matching network achieves the cross-modal alignment of the latent representations by solving the following entropic optimal transport (EOT) problem:

$$\widehat{T} = \arg\min_{T \in \Pi(\frac{1}{\|\hat{\mu}\|_1}\hat{\mu}, \gamma)} \underbrace{\langle D, T \rangle}_{\mathbb{E}_T[d(\bar{m}, \bar{a})]} + \lambda \underbrace{\langle T, \log T \rangle}_{\text{Entropy reg.}}, \qquad (4)$$

where $\langle \cdot, \cdot \rangle$ denotes the inner product of matrix, $\Pi(\frac{1}{\|\hat{\mu}\|_1}\hat{\mu}, \gamma) = \{T \geq 0 | T1_J = \frac{1}{\|\hat{\mu}\|_1}\hat{\mu}, T^\top 1_I = \gamma\}$ is the set of the doubly-stochastic matrix, whose marginals must be on the Simplex, i.e., $\frac{1}{\|\hat{\mu}\|_1}\hat{\mu} \in \Delta^{I-1}$ and $\gamma \in \Delta^{J-1}$. As shown in (7), the optimal solution $\widehat{T}$ is called optimal transport plan, which actually is the optimal distribution of latent representation pairs that minimizes the expectation of the distance $d(\bar{m}, \bar{a})$, whose marginal distributions are $\frac{1}{\|\hat{\mu}\|_1}\hat{\mu}$ and $\gamma$, respectively. Here, $\frac{1}{\|\hat{\mu}\|_1}\hat{\mu}$ determines the distribution of movie shots, and we set it as the normalized probabilities predicted by the movie shot selector. This setting ensures the predicted alignment result is consistent with the selection of movie shots. On the other hand, because each music shot is applied, we can simply set $\gamma$ to be uniform, i.e., $\gamma = \frac{1}{J}1_J$. Finally, the entropic regularizer of the OT plan improves the smoothness of the problem, whose significance is controlled by the hyperparameter $\lambda > 0$.

The EOT problem can be solved efficiently by the Sinkhorn-scaling algorithm shown in Algorithm 1, leading to the implementation of the Sinkhorn matching network with computational complexity $O(IJ)$. Note that the whole algorithmic process is differentiable for both $\widehat{T}$ and the distance matrix $D$ [27, 50], making the backpropagation applicable in the training phase. As a result, given $\widehat{T} = [\hat{t}_{ij}]$, we can select and align movie shots according to the music shots, i.e., $\hat{i} = \arg\max_{i \in \{1,...,I\}} \hat{t}_{ij}$ for $j = 1, ..., J$.

- **Remark.** It should be noted that compared with selecting and aligning movie shots based on the distance matrix $D$ (i.e., $\hat{i} = \arg\min_{i \in \{1,...,I\}} d(\bar{m}_i, \bar{a}_j)$ for $j = 1, ..., J$), the Sinkhorn matching network often provides better alignment results. In particular, without any constraint, the distance-based alignment may select the same movie shot to match with multiple music shots, which does harm to the diversity of the generated trailer and thus is undesired in practice. On the contrary, the doubly stochastic constraint encourages the optimal transport plan $\widehat{T}$ to achieve the one-one correspondence between the movie and music shots.

## 3.3 Inverse partial optimal transport framework

*3.3.1 The IPOT-based supervised learning paradigm.* Denote $\theta$ as the model parameters in the MLPs and the self- and cross-attention modules. When the dataset $\mathcal{D} = \{\mathcal{M}_n, \mathcal{V}_n, \mathcal{A}_n, T_n\}_{n=1}^N$ is available, we learn our model in a supervised way by solving the following inverse partial optimal transport (IPOT) problem:

$$\min_\theta \sum_{n=1}^N \underbrace{\text{KL}\left(\widehat{T}_n(\theta) \| \frac{1}{J_n}T_n\right)}_{\text{Supervision of Aligner}} + \delta \underbrace{\text{BCE}\left(\hat{\mu}_n(\theta), \mu_n\right)}_{\text{Supervision of Selector}}$$

$$s.t. \underbrace{\widehat{T}_n(\theta) = \arg\min_{T \in \Pi(\mu_n, \gamma_n)} \langle D_n(\theta), T \rangle + \lambda\langle T, \log T \rangle}_{\text{Entropic Partial Optimal Transport}}, \qquad (5)$$

$$\forall n = 1, ..., N.$$

As shown in (5), the IPOT problem is a bi-level optimization problem. In the upper-level problem, given each observed alignment matrix $T_n$, we take its normalized version $\frac{1}{J_n}T_n$ as the ground truth optimal transport plan between $\mathcal{M}_n$ and $\mathcal{A}_n$ and supervise the learning of our movie-music aligner. $\mu_n = T_n 1_{J_n}$ denotes the ground truth of movie shot selection, which supervises the learning of our movie shot selector. For each movie-music pair, the first term in the upper-level problem penalizes the KL-divergence between the predicted alignment result $\widehat{T}_n(\theta)$ and the ground truth $\frac{1}{J_n}T_n$. The second term penalizes the binary cross-entropy (BCE) loss between the predicted selection probabilities $\hat{\mu}_n(\theta)$ and the ground truth $\mu_n$. $\delta > 0$ controls the trade-off between the two terms.

The constraint of the upper-level problem corresponds to the lower-level optimization problem deriving the optimal transport plan. Compared with the EOT problem in (7), the lower-level problem in (5) takes the ground truth $\mu_n$ as the marginal distribution directly. Because of the sparsity of $\mu_n$, this problem is formulated as an entropic partial optimal transport problem — the rows of $\widehat{T}_n(\theta)$ corresponding to those unselected movie shots are set to be all-zeros so that we do not need to consider them during training. Such a strategy helps to decouple the learning objectives in the upper-level problem. In particular, while the BCE term focuses on learning the movie shot selector, by filtering out unselected movie shots, the KL-divergence term in the upper-level problem supervises the learning of the movie-music aligner for the selected movie shots and the music shots, which avoids the unnecessary mismatching with unselected movie shots.

*3.3.2 Learning algorithm.* This IPOT problem can be solved efficiently by a stochastic gradient descent (SGD) algorithm. Given

a batch of movie-music pairs, i.e., $\mathcal{B} \subset \mathcal{D}$, we first solve a set of entropic partial optimal transport problems and obtain the optimal transport plans for each movie-music pair, i.e., for $n \in \mathcal{B}$, we obtain $\widehat{T}_n(\theta) = \text{SinkhornNet}(D_n(\theta), \mu_n, \frac{1}{J_n}\mathbf{1}_{J_n}; \lambda)$. Then, we update the model parameters using SGD. Denote the objective function of the upper-level problem corresponding to the batch as $L(\theta)$. When computing the gradient of $L(\theta)$, we leverage the hypergradient method in [27, 51], i.e.,

$$\nabla_\theta L(\theta) = \sum_{n \in \mathcal{B}} \frac{\partial L(\theta)}{\partial D_n(\theta)} \frac{\partial D_n(\theta)}{\partial \theta} + \frac{\partial L(\theta)}{\partial \widehat{T}_n(\theta)} \frac{\partial \widehat{T}_n(\theta)}{\partial \theta}, \qquad (6)$$

in which the second term involves the hypergradient term $\frac{\partial \widehat{T}_n(\theta)}{\partial \theta}$, which requires us to unroll the Sinkhorn-scaling iterations in Algorithm 1. In general, this hypergradient term can be derived either by auto-differentiation [16, 53]. In our case, because $\widehat{T}_n(\theta)$ is the solution of an entropic optimal transport problem, this term can also be derived in a closed form. Please refer to Theorem 2 in [50] for more details. The remaining terms in (6) are derived by backpropagation.

### 3.4 Trailer generation pipeline

Given a well-trained model $\theta^*$, we can achieve music-guided movie trailer generation by an efficient pipeline. In particular, given a movie, we first resize it to 320p and then apply the video segmentation tool BaSSL [30] to obtain movie shots, i.e., $\mathcal{M} = \{m_i\}_{i=1}^I$. When a piece of music is provided, we first use the Ultimate Vocal Remover (UVR) tool to eliminate the vocal part, leaving only the background track, and then obtain music shots by the music segmentation tool Ruptures [43] method, i.e., $\mathcal{A} = \{a_j\}_{j=1}^J$. As aforementioned, both the movie and music shots are initially embedded by a pre-trained ImageBind [14].

By utilizing the well-trained movie shot selector and the latent representation model, we calculate the probability vector $\hat{\mu}(\theta^*)$ for movie shots and select the shots with $J$ highest probabilities to construct the trailer, i.e., $\mathcal{V} = \arg\text{ sort-}J_{i \in \{1,...,I'\}} \hat{\mu}_i$. Here, $I' = 0.9I$, which means that we only consider the first 90% of movie shots instead of all shots for spoiler prevention. After deriving the final latent representations of selected movie shots and music shots, i.e., $\bar{V} = [\bar{v}_j] \in \mathbb{R}^{J \times D}$ and $\bar{A} = [\bar{a}_j] \in \mathbb{R}^{J \times D}$, we infer the one-one correspondence between them by solving an EOT problem:

$$\widehat{T} = \arg\min_{T \in \Pi(\frac{1}{J}\mathbf{1}_J, \frac{1}{J}\mathbf{1}_J)} \langle D, T \rangle + \lambda \langle T, \log T \rangle, \qquad (7)$$

where the distance matrix $D = [d_{ij}] \in \mathbb{R}^{J \times J}$ contains the discrepancy between each selected movie shot and each music shot. In this study, we define its elements as

$$d_{ij} = \underbrace{\|\bar{v}_i - \bar{a}_j\|_2^2}_{\text{Semantic dis.}} + \eta \underbrace{|\tau_i^m - \tau_j^a|}_{\text{Temporal dis.}}, \ \forall i, j = 1, ..., J. \qquad (8)$$

Here, for the $i$-th selected movie shot and the $j$-th music shot, the first term in (8) indicates the semantic discrepancy between their latent representations, while the second term in (8) indicates their temporal discrepancy, where $\tau_i^m$ and $\tau_j^a$ denote the lengths of the two shots. Typically, it is easy to synchronize the two shots when $|\tau_i^m - \tau_j^a|$ is small. The hyperparameter $\eta > 0$ achieves the trade-off between the two terms. Note that, in the training phase, we do not consider the temporal discrepancy when constructing the

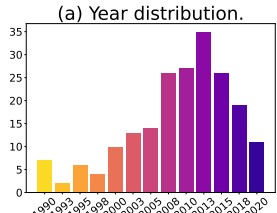
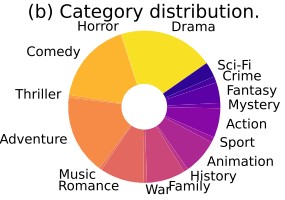

**Figure 2: Visualization of the publication year distribution and the category proportions of movies in CMTD.**

$D_n(\theta)$'s in (5) because the aligned shots in our training data have the same duration while those unaligned shots often have different lengths. Introducing the temporal discrepancy would oversimplify the learning task and weaken the supervision on our model.

After inferring the aligned shot pairs based on $\widehat{T}$, we engage in post-processing the aligned movie shots to adapt the duration of the music shots. For each music shot, when the corresponding movie shot exceeds its duration, we cut the movie shot to match its duration. When the corresponding movie shot falls short on length, we extend the movie shot by incorporating one or more adjacent movie shots based on their probabilities (i.e., $\hat{\mu}(\theta^*)$). Finally, we concatenate the post-processed movie shots and take the music as the soundtrack, composing the ultimate movie trailer.

## 4 THE CMTD DATASET FOR TRAINING

Implementing our IPOT learning framework needs a movie-trailer dataset with detailed annotations (e.g., the segmentation and alignment information). Unfortunately, existing datasets, e.g., Large-Scale Movie and Trailer Dataset (LSMTD) [18], Trailer Momont Detection Dataset (TMDD) [47] and Movie Highlight Detection Dataset (MovieLights) [11], are non-public and fail to meet our requirement. The movies and trailers in LSMTD are not paired. Although the movies and trailers in TMDD and MovieLights are paired, each movie is only associated with a single trailer (and its music). In the music-guided trailer generation task, we expect that each movie corresponds to multiple trailers, which helps suppress the risk of over-fitting. The above problems motivate us to build our Comprehensive Movie-Trailer Dataset, called CMTD for short.

### 4.1 Data collection and annotation

As shown in Figure 2, CMTD contains 208 movies and 406 trailers. These movies and trailers have sufficient richness and diversity in content and year, which are categorized into 18 classes based on their tags at IMDB.[2] Each movie corresponds to one to six trailers, roughly two trailers per movie on average. The average duration per movie and trailer is 1.91 hours and 2.17 minutes, respectively. We apply BaSSL [30] to segment each movie/trailer into shots and aggregate adjacency shots in the scene level. The average shot number and scene number per movie are 1909 and 57, respectively.

To obtain the alignment matrix automatically with high accuracy, for each movie-trailer pair, we first obtain frame-level visual embeddings for the movie and the trailer by ImageBind [14]. Based on

[2]http://www.imdb.com/

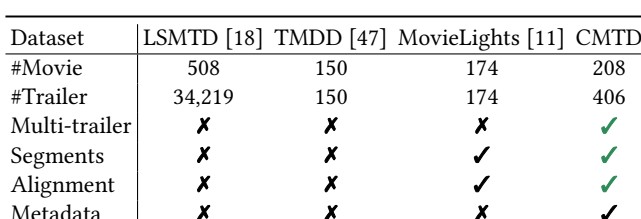

Figure 3: An illustration of the annotations associated with the movie "The Great Gatsby" in CMTD.

| Dataset | LSMTD [18] | TMDD [47] | MovieLights [11] | CMTD |
|---|---|---|---|---|
| #Movie | 508 | 150 | 174 | 208 |
| #Trailer | 34,219 | 150 | 174 | 406 |
| Multi-trailer | ✗ | ✗ | ✗ | ✓ |
| Segments | ✗ | ✗ | ✓ | ✓ |
| Alignment | ✗ | ✗ | ✓ | ✓ |
| Metadata | ✗ | ✗ | ✗ | ✓ |

Table 1: A comparison for various datasets. We use green ticks to mark the information used for trailer generation.

the embeddings, we then apply Faiss [8, 20] to efficiently compute the visual similarity between the trailer's frames and the movie's. For each trailer frame, we can get the top-4 movie frames that most closely resemble it. For a trailer shot having $K$ frames, we can select $4K$ most similar movie shots — each trailer frame is matched with four movie shots where the corresponding top-4 movie frames belong to. Among the $4K$ movie shots, the shot with the highest number of occurrences is annotated as the correspondence of the trailer shot. Applying this annotation method to all trailer shots, we can calculate the alignment matrix $T$ between the movie and trailer shots. By random sampling and manual verification, we confirm the reliability of this annotation method.

Besides the shot-level alignment information, CMTD also provides abundant auxiliary information as metadata, including subtitles, synopsis, turning points annotations, and so on. In particular, for each movie, we collect its subtitle from Subscene[3] and collect its synopsis from the movie's Wikipedia page. Based on the synopsis, we further annotate five turning points (i.e., the key moments in the storyline) that define the narrative structure of the movie [34, 35]. Figure 3 illustrates the annotations associated with a movie, and Table 1 shows the comparison among different datasets. Note that, although we just apply the segment and alignment information in the trailer generation task (for a fair comparison with baseline methods), the metadata in CMTD can support more applications and thus contribute to promoting more studies of video understanding.

## 4.2 Data release plan and its social impacts

We plan to make our CMTD dataset public. To achieve a trade-off between the acceleration of research and the protection of intellectual property, we are considering the following strategies.

[3]http://www.subscene.com/

- We plan to develop a license agreement that further stipulates the use scope and limitations, including prohibiting redistribution, commercial use, modification, etc., to ensure the dataset is used only for non-commercial research and academic purposes. Before accessing our data, each user is required to submit a signed application form and provide his/her education email, promising to obey the agreement.
- For all movies, trailers, and music, we plan to release their embeddings and annotations, including those metadata, such that if the users can access the raw videos, they can easily segment and align the videos based on the annotations. Additionally, for the convenience of research, we also consider releasing movies and trailers with extremely low resolutions and/or watermarks, preventing them from being used for other purposes except research.

We expect CMTD to be the first publicly available movie-trailer dataset, advancing the academic field of video understanding and triggering more interesting and significant research work.

## 5 EXPERIMENTS

To demonstrate the effectiveness of our IPOT-based trailer generator, we compare it with state-of-the-art methods through both objective and subjective evaluations. **The code, demo videos, and more experimental results are in supplementary file.**

## 5.1 Implementation Details

*5.1.1 Baselines.* We take state-of-the-art trailer generation methods as baselines, including V2T [19], M2T [42], and PPBVAM [55]. When evaluating the movie shot selector learned by our method, we also compare it with three state-of-the-art video summarization methods (*i.e.*, VASNet [9], CLIP-It [32], and OTVS [49]) and one commercial video summarization software Muvee [12]. For learning-based methods, including ours, we select 200 movies from CMTD for training and apply the remaining eight movies for evaluation. Note that, because most of the baselines only release trailers generated from the eight movies rather than their code, we select the eight movies for a fair comparison.

*5.1.2 Evaluation Metrics.* For the ground truth trailer and the generated one, we can find the indices of their shots in the corresponding movie and construct two index sequences, denoted as $A = \{\alpha_1, ..., \alpha_I\}$ and $B = \{\beta_1, ..., \beta_I\}$. Therefore, when evaluating our movie shot selector, we take three commonly used metrics:

| Category | Method | Movie shot selection | | | | | | | | | Movie-music alignment | | | |
|---|---|---|---|---|---|---|---|---|---|---|---|---|---|---|
| | | P@1↑ | P@3↑ | P@5↑ | R@1↑ | R@3↑ | R@5↑ | F1@1↑ | F1@3↑ | F1@5↑ | P@1↑ | R@1↑ | F1@1↑ | KL↓ |
| Video Summary | VASNet [9] | 0.0237 | 0.0725 | 0.1102 | 0.0343 | 0.1096 | 0.1698 | 0.0277 | 0.0861 | 0.1317 | — | — | — | — |
| | Muvee [12] | **0.2130** | **0.3245** | **0.3452** | 0.0414 | 0.0612 | 0.0690 | 0.0640 | 0.0949 | 0.1059 | — | — | — | — |
| | CLIP-It [32] | 0.0302 | 0.0863 | 0.1468 | 0.0527 | 0.1409 | 0.2429 | 0.0378 | 0.1054 | 0.1801 | — | — | — | — |
| | OTVS [49] | 0.0637 | 0.1398 | 0.1821 | 0.0941 | _0.2157_ | 0.2864 | 0.0746 | _0.1669_ | _0.2193_ | — | — | — | — |
| Trailer Generation | M2T [42] | 0.0229 | 0.0347 | 0.0444 | 0.0188 | 0.0273 | 0.0362 | 0.0193 | 0.0285 | 0.0371 | 0.0028 | 0.0031 | 0.0029 | 2417.71 |
| | V2T [19] | 0.0787 | 0.1397 | 0.2031 | 0.0396 | 0.0693 | 0.1035 | 0.0508 | 0.0891 | 0.1322 | _0.0028_ | _0.0031_ | _0.0029_ | 1855.59 |
| | PPBVAM [55] | 0.0687 | 0.1339 | 0.1862 | _0.1003_ | 0.2000 | 0.2729 | _0.0781_ | 0.1537 | 0.2117 | 0.0019 | 0.0022 | 0.0020 | 2871.03 |
| | IPOT (Ours) | _0.1098_ | _0.2248_ | _0.3064_ | **0.1234** | **0.2536** | **0.3446** | **0.1161** | **0.2381** | **0.3240** | **0.0075** | **0.0081** | **0.0078** | **1696.42** |

**Table 2: Comparisons on movie shot selection. We bold the best results and underline the second-best results.**

Top-$K$ Precision, Recall, and F1-Score, i.e., $P@K = |A \cap_K B| / |A|$, $R@K = |A \cap_K B| / |B|$, and $F1@K = \frac{2P@K \cdot R@K}{P@K + R@K}$, where $| \cdot |$ is the cardinality of set, $K = 1, 3, 5$, and $A \cap_K B = \{i | |\alpha_i - \beta_i| \leq K - 1\}$ counts the number of shot pairs with close enough indices.

For each trailer generator, we use P@1, R@1, F1@1, and KL-divergence between the estimated and observed alignment matrices to quantitatively evaluate the alignment between movie shots and audio shots. Additionally, the statistics of trailer shots is applied to evaluate the quality of generated trailers as well — we record the number and average duration of trailer shots generated by each method and compare these values with those of official trailer shots. Besides objective measurements, we also evaluate different methods through subjective user study.

*5.1.3 Model and hyperparameter settings.* The self-attention and cross-attention modules are implemented as Transformer encoders, each of which has one layer and two attention heads. There are linear layers both before and after the two types of attention modules. When training our model, we apply Adam [21] with $\beta_1 = 0.9$ and $\beta_2 = 0.999$. The learning rate is 1e-5 and the training epoch is 500.

## 5.2 Quantitative and Qualitative Comparisons

Table 2 shows the performance of various methods on movie shot selection. We can see that our IPOT-based method achieves the best performance on most measurements. Especially in terms of F1-Scores, our method works best in all three settings. These results demonstrate the superiority of our method on movie shot selection — it is more likely to select the movie shots that are used in official trailers. In the aspect of movie-music shot alignment, we mainly compare our method with other trailer generators. The results in Table 2 show that our method can achieve the highest precison, recall, and F1-score and the lowest KL-divergence, which means that the alignment achieved by the OT plan matches better with the ground truth than other methods. Figure 4 provides an example comparing the generated trailers of different methods with the official one, which visualizes the advantage of our method.

Table 3 shows the comparison on the number and average duration of trailer shots generated by various methods. The number of shots in different trailers is distinct. Some methods choose a very small number of shots in a trailer, such as Muvee, making these methods achieve high precision but low recall. On the contrary, some methods choose a massive number of trailer shots, such as PPBVAM [55], making them dominant in recall. Our method is

| Methods | Test movie-1 | | Test movie-6 | |
|---|---|---|---|---|
| | Duration (s) | #Shot | Duration (s) | #Shot |
| Official Trailer | $1.95_{\pm 1.82}$ | 77 | $2.35_{\pm 2.91}$ | 63 |
| PPBVAM [55] | $1.12_{\pm 0.39}$ | 163 | $1.26_{\pm 0.46}$ | 131 |
| Muvee [12] | $7.72_{\pm 8.73}$ | 24 | $42.69_{\pm 33.83}$ | 4 |
| V2T [19] | $4.06_{\pm 5.72}$ | 44 | $2.83_{\pm 2.53}$ | 58 |
| M2T [42] | $1.71_{\pm 0.83}$ | 89 | $1.72_{\pm 0.75}$ | 89 |
| IPOT (Ours) | $2.03_{\pm 2.00}$ | 74 | $2.01_{\pm 1.73}$ | 58 |

**Table 3: Comparisons on trailer shot number and duration.**

more balanced, whose number and duration of trailer shots are close to those of official trailer shots, so that it achieves the best performance on F1-scores.

## 5.3 Subjective User Study

Besides objective evaluation, we evaluate our method as well as the baselines (i.e., V2T, M2T, PPBVAM) through subjective user studies, comparing their user scores with those of official trailers (RT). Following the work in [19, 55], we propose to compare different trailers in the following five aspects:

- **Character**: How does the trailer include close-up shots of the main characters in the movie?
- **Rhythm**: How well does the montage match the rhythm of the background music?
- **Attractiveness**: How attractive is the trailer? How much are you impressed by this trailer?
- **Appropriateness**: How close is the trailer to a real trailer?
- **Interest**: How interested do you become in watching this movie after watching the trailer?

All trailers are processed to the same resolution (320×240). Given the trailers generated by different methods, we establish a website and invite 25 volunteers (7 females and 18 males) to watch them, in which the names of the methods are anonymous and the order of the trailers on the website is random. For each movie, a volunteer scores the corresponding generated trailers from one (the lowest) to seven (the highest) in each of the above five aspects, where the score of the official trailer is set to be seven by default. Figure 5 shows the results of various methods. On average, our method consistently outperforms the three baselines in the five aspects.

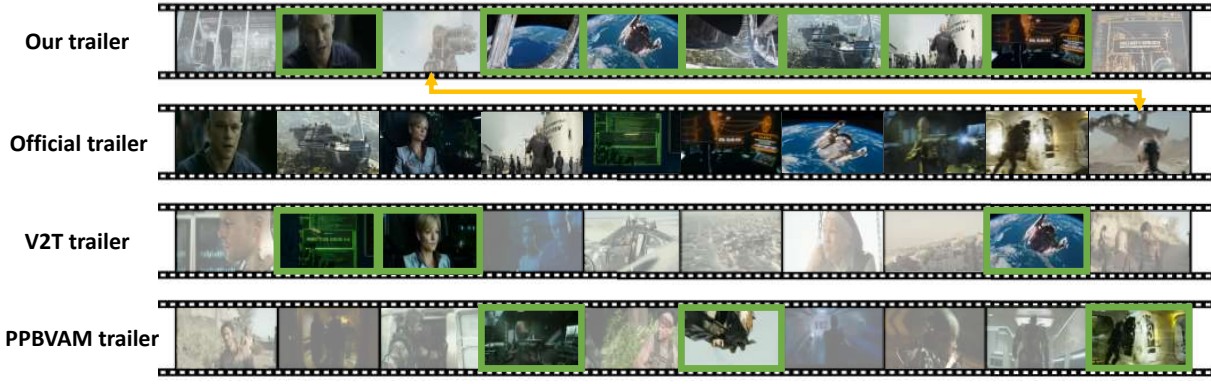

**Figure 4: Comparison between some generated trailer shots and the official trailer shots of the movie "Elysium" based on their appearance order. For each generated trailer, their correctly selected shots are marked with green boxes. The selected shot of our trailer connected to the shot in the official trailer by a yellow arrow means that they belong to the same scene.**

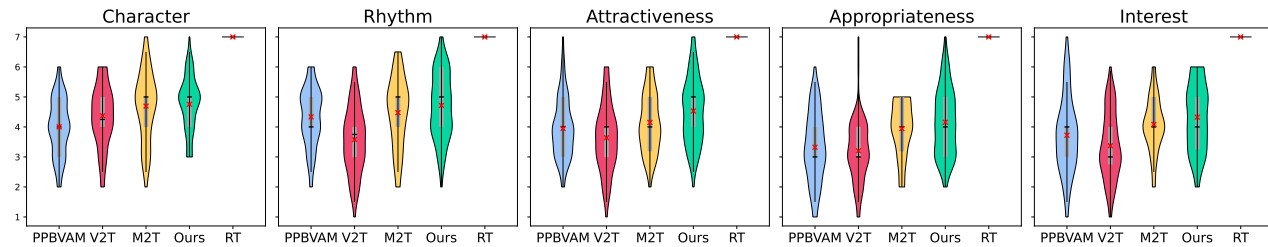

**Figure 5: The violin plot of scores for various methods in user studies. The red crosses are means and the black bars are medians.**

| Setting | Selection | | | Alignment | |
|---|---|---|---|---|---|
| | F1@1↑ | F1@3↑ | F1@5↑ | F1@1↑ | KL↓ |
| -w/o SA | 0.0280 | 0.0815 | 0.1015 | 0.0011 | 1706.50 |
| -w/o CA | 0.0317 | 0.0907 | 0.1454 | 0.0020 | 1704.48 |
| -w/o PartialOT | 0.1077 | 0.1536 | 0.1976 | 0.0036 | 1702.47 |
| **Proposed** | **0.1129** | **0.2178** | **0.3240** | **0.0078** | **1696.42** |

**Table 4: Ablation study on the model components.**

Note that, because the quality of the movie trailer is finally evaluated by the audience in practice, which is highly subjective, the above user study is necessary and can provide complementary information compared to the objective measurements. For example, the F1-score of M2T in movie shot selection is very low, but its scores in the user study are better than V2T and PPBVAM. This phenomenon implies that it may select relatively reasonable shots that are not used in official trailers.

### 5.4 Ablation Study

Table 4 displays the results of some ablation experiments, demonstrating the significance of different model components. Removal of the self-attention (SA) mechanism from the framework results in a significant degradation in both selection and alignment performance. This may be attributed to SA's role in establishing temporal relationships among shots and enabling the integration of

contextual information. Eliminating the cross-attention (CA) mechanism disrupts the semantic interaction between the two modalities, making their alignment more challenging and leading to a notable decrease in performance. In addition, when training the model without the partial OT mechanism, i.e., replacing the $\mu_n$ with $\hat{\mu}_n$ in the lower-level problem of (5), the model performance also degrades because of introducing unnecessary uncertainty.

## 6 CONCLUSION AND FUTURE WORK

In this work, we propose an inverse partial optimal transport (IPOT) framework for music-guided trailer generation and build a comprehensive movie-trailer dataset to support the learning of the trailer generator. The proposed trailer generator consists of a music-guided movie shot selector and a movie-music shot aligner, which can be learned effectively by a bi-level optimization strategy. Experiments demonstrate that our IPOT-based method outperforms state-of-the-art trailer generation and video summarization methods on both objective and subjective evaluation measurements.

Currently, the generated trailers are still incomparable to the human-edited trailers in quality, as shown in Figure 5, which are far from practical applications. In the future, we plan to further enlarge out CMTD dataset, collecting more movies and trailers with metadata to support the learning of the model. In addition, we would like to utilize more side information to learn the model, including but not limited to subtitles, turning points, and synopsis.

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
