# OpenReview forum: "An Inverse Partial Optimal Transport Framework for Music-guided Trailer Generation"
_acmmm.org/ACMMM/2024/Conference — MM2024 Poster_

### Official Review · Reviewer_rM1L · 2024-05-24

**Rating:** 4
**Confidence:** 2

**Summary:**

The paper presents a novel approach to music-guided movie trailer generation using an Inverse Partial Optimal Transport (IPOT) framework. The proposed method formulates the trailer generation task as selecting and sorting key movie shots based on audio shots, leveraging a two encoder to derive latent representations and an attention-assisted Sinkhorn matching network to align these representations. The model is trained using a bi-level optimization strategy, solving an inverse partial optimal transport problem. A new dataset, CMTD, is introduced to support the learning and evaluation of the model.

**Strengths:**

1.The use of the IPOT framework for trailer generation is innovative, combining standard machine learning techniques in a new way.

2.The introduction of the CMTD dataset, with detailed annotations and metadata, provides a valuable resource for the research community.

3.The methodology is technically sound, supported by detailed equations, algorithms, and a clear explanation of the optimization process.

**Limitations:**

1.While the method shows strong performance on the CMTD dataset, its generalizability to other datasets or types of video content is not/could not be thoroughly explored.

2.The method relies heavily on the availability of detailed annotations and metadata, which may not always be accessible or feasible to obtain for all datasets.

3.require substantial computational resources, potentially limiting the method's applicability in resource-constrained environments.

4.The comparisons with recent state-of-the-art methods could be expanded to provide a more comprehensive evaluation of the proposed method's relative performance

**Suitability:**

3

---

### Official Review · Reviewer_UmkS · 2024-05-24

**Rating:** 4
**Confidence:** 2

**Summary:**

This paper proposes a novel movie trailer generation by selecting and sorting key movie
shots based on audio shots. The authors also provide a corresponding dataset for further research.

**Strengths:**

1. The given data set has some new properties that will facilitate further research.
2. The proposed method outperforms existing methods in terms of both numerical results and user study.

**Limitations:**

1. Why are only the music shots of the trailer shots used in the pipeline? It seems to me more reasonable to use all music shots along with movie shots to get the alignment information.
2. I am not sure how we use the IPOT model for testing. Is the music shot of trailers needed in the test step?

**Suitability:**

3

---

### Official Review · Reviewer_o9S2 · 2024-05-24

**Rating:** 4
**Confidence:** 3

**Summary:**

The paper "An Inverse Partial Optimal Transport Framework for Music-guided Movie Trailer Generation" presents a novel approach for automatically generating movie trailers guided by background music. The proposed framework, termed Inverse Partial Optimal Transport (IPOT), formulates trailer generation as a problem of selecting and ordering key movie shots based on corresponding audio segments. This involves matching latent representations across visual and acoustic modalities. The IPOT framework employs a two-tower encoder to derive these latent representations and an attention-assisted Sinkhorn matching network to parameterize the grounding distances between the visual and acoustic shots. The model is trained on a newly constructed dataset of real-world movies and their trailers, labeled with abundant information, called CMTD. The authors demonstrate that their method outperforms state-of-the-art techniques in both subjective visual effects and objective quantitative measurements.

**Strengths:**

The paper introduces a unique approach to trailer generation by leveraging inverse partial optimal transport, which is a significant contribution to the field. The IPOT framework represents a novel application of optimal transport techniques to the problem of matching audio and visual modalities for trailer generation.

The IPOT framework is well-structured, integrating multiple components like the two-tower encoder and attention-assisted Sinkhorn matching network effectively. This comprehensive approach ensures robust performance in aligning and selecting key shots from movies based on the audio guidance provided by the background music.

The creation of the CMTD dataset, with its rich annotations and metadata, is a valuable resource for future research in video understanding and trailer generation. This dataset not only supports the training and evaluation of the proposed model but also provides a benchmark for future studies in this domain.

The authors provide thorough experimental validation, demonstrating the superiority of their method over existing state-of-the-art techniques in both objective metrics and subjective user studies. The extensive experiments and user studies underline the practical applicability and effectiveness of the proposed framework.

**Limitations:**

The method essentially performs a similarity search between music and video segments. This approach has limited generalization ability, especially in unsupervised scenarios where finding a complete set of ground truth music shots corresponding to all movie shots may not be feasible. The reliance on supervised learning constrains the model’s applicability in real-world situations where labeled data is scarce.

The generalization ability of the method is weak, as it relies heavily on supervised learning. In real-world applications, data is often unlabeled, and it might be challenging to obtain ground truth annotations for all possible movie and music shot combinations. This limitation reduces the method’s effectiveness in diverse and unsupervised environments.

Generative models might offer better practical applicability and generalization ability. They can potentially handle unsupervised data more effectively and provide more robust trailer generation capabilities across diverse movie genres and styles. Exploring the integration of generative models could enhance the practical utility and scalability of the proposed approach.

**Suitability:**

3

---

### Meta-Review · Area_Chair_eynH · 2024-07-03

**Recommendation:** Accept (Poster)
**Confidence:** 5

**Metareview:**

The reviewers propose the acceptance of the paper. The authors can find the detailed review. There are pros and cons, as usual. The impact of the results would fit a poster presentation.